# Re-Evaluation of the Cross-Reactions of the Antibody against the Causative Agent for Paracoccidioidomycosis Ceti; *Paracoccidioides ceti* and the Related Fungal Species

**DOI:** 10.3390/microorganisms11102428

**Published:** 2023-09-28

**Authors:** Hikaru Kanegae, Igor Massahiro de Souza Suguiura, Rentaro Tashiro, Toshihiro Konno, De-Xing Hou, Ayako Sano, Takeshi Eto, Keiichi Ueda, Md. Amzad Hossain

**Affiliations:** 1The United Graduate School of Agricultural Sciences, Kagoshima University, 1-21-24 Korimoto, Kagoshima 890-0065, Japan; k6695082@kadai.jp (H.K.);; 2Department of Pathological Sciences, State University of Londrina, P.O. Box 10011, Londrina 86057-970, PR, Brazil; 3Paraná State Secretariat of Health, Department of Health Surveillance, 17th Health Region, Alameda Miguel Blasi, 76-Centro, Londrina 86010-070, PR, Brazil; 4Graduate School of Agriculture, University of the Ryukyus, Sembaru 1, Nishihara-Cho, Nakagami-Gun 903-0213, Japan; 5Faculty of Agriculture, University of the Ryukyus, Sembaru 1, Nishihara-Cho, Nakagami-Gun 903-0213, Japan; teto@agr.u-ryukyu.ac.jp; 6Faculty of Agriculture, Kagoshima University, 1-21-24 Korimoto, Kagoshima 890-0065, Japan; 7Okinawa Churashima Foundation, Aza Ishikawa 888, Motobu-Cho, Kunigami-Gun 905-0206, Japan

**Keywords:** antibody, highly pathogenic fungal infections, paracoccidioidomycosis ceti, serological cross-reaction

## Abstract

Paracoccidioidomycosis ceti (PCM-C) is a chronic granulomatous keloidal dermatitis in cetaceans that has been reported worldwide and is caused by *Paracoccidioides ceti*. Serological cross-reactions among highly pathogenic fungal infections and related diseases have been reported. However, the true cross-reaction of antibodies against *P. ceti* has remained unknown due to the use of positive control sera from infected dolphins. This study aimed to re-evaluate antibodies from mechanically dislodged fungal cells in the infected tissue of a PCM-C case and demonstrate the actual cross-reaction. The results revealed a limited cross-reaction between PCM-C and paracoccidioidomycosis, while the antibodies did not react with other pathogens such as *Coccidioides posadasii*, *Histoplasama capsulatum*, and *Arthrographis kalrae*. Thus, the method for evaluation of the antibody against PCM-C is reliable, and there is potential for epidemiological study.

## 1. Introduction

Paracoccidioidomycosis ceti (PCM-C) is an intractable mycosis characterized by chronic granulomatous keloidal dermatitis in cetaceans that has been reported worldwide [1]. It is treated as a zoonosis based on a human case of a dolphin trainer who handled an infected animal [2]. The causative agent for PCM-C was renamed *Paracoccidioides ceti* by Vilela et al. in 2023 [3].

The cetacean species known to be hosts for PCM-C include the Atlantic bottlenose dolphin (*Tursiops truncatus*), Indo-Pacific bottlenose dolphin (*T. aduncus*), Pacific white-sided dolphin (*Lagenorhynchus obliquidens*), estuarine dolphin (*Sotalia guianensis*, also known as the costero dolphin), Indian Ocean humpback dolphin (*Sousa plumbea*) [1], and false killer whale (*Pseudorca crassidens*) [4].

Diagnosis of PCM-C is based on clinical symptoms and the detection of typical round yeast cells arranged in chains or producing multiple buddings. Molecular biological and serological data serve as auxiliary methods for diagnosis [1,5,6]. These methods have aided in the epidemiology of PCM-C in nursing and wild dolphins [7,8]. However, the potential for serological cross-reactions with closely related fungal species belonging to the order Onygenales and the family Ajellomycetaceae cannot be ignored, as there are case reports of histoplasmosis, coccidioidomycosis, adiasporomycosis, and *Chrysosporium* spp. infections in Asian countries [8,9]. Additionally, the effects of serological cross-reactions caused by *Arthrographis karlae* infections should not be disregarded [6]. However, the aforementioned serological cross-reactions have not been evaluated using pure antibodies against *P. ceti*. Interestingly, we found independent seropositivity against *P. ceti* and *C. posadasii* in wild dolphins [8], which suggests that there is no serological cross-reaction between PCM-C and coccidioidomycosis.

The objective of the present study is to investigate the cross-reactions of pure antibodies against the causative agent of paracoccidioidomycosis ceti (*Paracoccidioides ceti*) and those of related fungal species associated with the causative agents of paracoccidioidomycosis, coccidioidomycosis, histoplasmosis, and *A. karlae* infection.

## 2. Materials and Methods

### 2.1. Pure Antibody against Paracoccidioidomycosis Ceti

Anti-*P. ceti* sera from a rabbit were produced by a company (Eurofins Genomics K. K., Tokyo, Japan) using the fungal cells derived from the first case of PCM-C in Japan [10]. (Figure 1). A mass of approximately 10 × 10 × 10 mm^3^ containing numerous fungal cells was taken from a 10% formalin-fixed dermal sample using surgical scalpels and tweezers and cut with scissors into numerous small pieces approximately 2 × 2 × 2 mm^3^ in size. The several small pieces prepared were mixed with 10 mL of sterile phosphate-buffered saline (PBS, Fujifilm Wako Pure Chemical Co., Osaka, Osaka, Japan; Wako), poured into a 20 mL volume-sized glass homogenizer, and well grinded by hand until the masses became a milky liquid. The liquid containing the fungal cells was diluted approximately 10^8^ cells/mL with 40 mL of sterile PBS using a 50 mL sized plastic centrifuge tube (As one Co., Osaka, Oasaka, Japan; As one), washed 3 times with sterile PBS, centrifuged at 1710× *g* for 5 min, suspended in 40 mL of sterile PBS, and left standing for 10 min. Twenty milliliters of the supernatant of the fungal solution was gently collected using a plastic 10 mL pipette, and the concentration of the fungal cells was adjusted to 10^7^ cells/mL. The fungal cell solution consisted of round yeast cells with a small debris of cetacean tissue, as shown in Figure 2. The procedure was performed in accordance with the company’s policy and the animal ethics of Eurofins Genomics Inc. (Eurofins Genomics Inc., Ota, Tokyo, Japan). Briefly, 1 mL of the fungal solution (10^7^ cells/mL) was injected into a female adult rabbit with a body weight of approximately 5 kg intravenously once a week for six weeks, and the initial (before immunization) and final (after six times) sera containing the antibody were obtained.

Anti-*P. brasiliensis* serum from a rabbit was supplied by Dr. Eiko Nakagawa Itano (Londrina, Prana, Brazil). Anti-*Coccidioides* spp. serum was selected from our previous report on a Dall’s porpoise [8]. In addition, we confirmed that the serum reacted against *C. posadasii* fungal cells independently during the present research. Anti-*H. capsulatum* antibody was obtained from pooled sera from five mice experimentally infected with IFM 41329 [5]. Anti-*A. karlae* antibody was also obtained from pooled sera from five mice experimentally infected with IFM 55165 [6]. The list of the antibodies used as positive control are shown in Table 1.

### 2.2. Fungal Cells Used as Antigens

The yeast cells (strain SUM, GenBank Accession No. AB811031) of *P. cetii* (derived from the skin lesion of the first Japanese case of PCM-C in a bottlenose dolphin) [10] embedded in paraffin blocks were used as antigens for PCM-C [4,5,6,7,8]. The cells from various stages of *C. posadasii* [11] (isolate IFM 4935, from a Japanese patient, with confirmation via genetic analysis [12]) in murine pulmonary tissue embedded in paraffin blocks [13] were used as antigens for coccidioidomycosis [5,6,8]. Briefly, the paraffin-embedded tissue samples were cut into pieces with a thickness of 8 µm, placed on poly-L-lysine-coated glass slides (S7441; Matsunami Glass Ind., Ltd., Kishiwada, Osaka, Japan; Matsunami), and deparaffinized, as previously reported [4,5,6,7,8].

The yeast cells of *P. brasiliensis* strain Pb-18, isolated from a human patient (IFM 41621), corresponding to *P. brasiliensis sensu stricto* [6], that of *H. capsulatum* IMT/HC12 [14] isolated from a Peruvian patient, and *A. karlae* IFM 55165 isolated from a cutaneous lesion of a cat [15] were cultured on 1% yeast extract (Becton, Dickinson and Company, Sparks, MD, USA; BD) and 2% dextrose (Wako) added to brain heart infusion agar (BD) slants at 35 °C for 5 days. The slants were fixed with 70% ethanol for 48 h, washed 3 times with distilled water with a centrifuge at 1710× *g* for 5 min, suspended in distilled water for approximately 10% of the volume, and stored in a refrigerator at 4 °C.

The cultured fungal cells were adjusted to the concentration at 10^7^/mL and suspended in distilled water containing 0.01% of ovalbumin (618-431-3; Wako). One microliter volume of the yeast cells of *H. capsulatum*, *P. brasiliensis*, and *A. kalrae* (Table 1) was spotted on poly L-lysine-coated glass slides in the 15 × 15 mm^2^ area. After air drying, the spotted place was encircled with waterproof ink (liquid blocker, Super pap pen; Daido Sangyo Co., Ltd., Ota, Tokyo, Japan). The slide glass was covered with a drop of methanol, approximately 200 μL for fixation, kept until the disappearance of the liquid, and the 200 μL of 0.1% ovalbumin was placed. Then, 200 μL of methanol was placed on the slide glass for re-fixation. After the disappearance of methanol, the slides were kept for immune reactions. The list of the antigens used as positive control are shown in Table 2.

### 2.3. Detection of the Antibodies via Immune Reactions

After being washed three times with phosphate-buffered saline (PBS; 167-14491; Wako), the deparaffinized samples and fixed fungal cells were blocked with 100 µL of 5% skim milk (19810605; Wako) dissolved in PBS (SM-PBS) for 15 min at room temperature. After discarding the SM-PBS, 100 µL of serum samples at 1000-fold dilution in SM-PBS (used as the primary antibody) was added to the tissues or fungal cells in a moistened box. SM-PBS without the primary antibody was used as a negative control. The samples were incubated at 4 °C for 16 h and washed three times with PBS. 

The positive sera derived from a rabbit were reacted with goat anti-rabbit IgG-H&L (HRP) (ab6721; Abcam, Cambridge, Cambridgeshire, UK; Abcam), those from mice were incubated with horseradish peroxidase-conjugated rabbit polyclonal anti-mouse immunoglobulin G antibody (ab97046; Abcam), and those from cetacean samples were then incubated with horseradish peroxidase-conjugated rabbit polyclonal anti-dolphin immunoglobulin G antibody (HRP-AD; 100 µL; ab112789; Abcam), and diluted 500-fold in PBS for 30 min at room temperature.

Then, the samples were washed three times with PBS and irradiated with 3,3′-diaminobenzidine (Histofine SAB-PO(M) Kit; Nichirei Biosciences, Chuou, Tokyo, Japan) for 10 min at room temperature, following the manufacturer’s instructions. Subsequently, the samples were stained with hematoxylin (131-0966; Wako), sealed with Canadian balsam (192-16,301; Wako), and observed under an optical microscope (Labophoto; Nikon, Minato, Tokyo, Japan). Samples were considered to test positive when fungal cell walls stained brown [4,5,6,7,8] at 1000-fold dilution in the present study.

## 3. Results and Discussion

### 3.1. Immune Staining

The fungal cell samples used as antigens were properly reacted with the positive antibodies. Figure 3 demonstrates the positive and negative immune reactions.

### 3.2. Immune Reactions

The fungal cells utilized as antigens for *P. ceti*, *C. posadasii*, *P. brasiliensis*, *H. capsulatum*, and *A. karlae* exhibited proper reactions with the positive sera. The anti-*P. ceti* and *P. brasiliensis* antibodies displayed positive reactions against *P. ceti* and *P. brasiliensis*. The anti-*Coccidioides* spp. serum, which was selected from our previous report on a Dall’s porpoise [8], independently reacted positively against *C. posadasii*. Pooled murine sera containing anti-*H. capsulatum* and anti-*A. karlae* antibodies demonstrated reactions with *P. ceti*, *C. posadasii*, *P. brasiliensis*, *H. capsulatum*, and *A. karlae* (Table 3).

### 3.3. Discussion

The present study established that the serological cross-reaction was limited between paracoccidioidomycosis and PCM-C. Similarly, Minakawa et al. [7] noted the absence of human paracoccidioidomycosis, blastomycosis, and coccidioidomycosis in Japan when analyzing the seroprevalence of *P. ceti* in Japanese aquaria. However, false seropositivity against *P. ceti* fungal cells caused by cross-reactions with *Coccidioides* spp., *H. capsulatum*, and *Arthrographis kalrae* detected in the sera from infected dolphins is still unknown [5,6]. 

Based on the present study, the survey on seroprevalence against *P. ceti* in dolphins [7,8] was considered reliable. In other words, the present study hypothesized that the seropositivity against *Coccidioides* spp., *H. capsulatum*, and *A. kalrae* in sera derived from PCM-C infected dolphins was due to exposure to the fungal pathogens. In fact, we have already detected independent exposures to *P. ceti* and *Coccidioides* spp. in a survey of wild dolphins living in the subarctic area [8].

It was certain that the sera derived from a wild dolphin used as a positive control against *C. posadasii* fungal cells [8] exhibited a definite reaction since the habitat of the dolphin had a lower incidence of fungal pathogens in the seawater [16]. The independent positivity against *C. posadasii* without any cross-reaction against *H. capsulatum* and *A. kalrae* observed in the present study was a natural occurrence.

Previously, we considered that the members of the order Onygenales and the family Ajellomycetaceae, including *Blastomyces dermatitidis*, *Histoplasma capsulatum*, and *Paracoccidioides* spp., and the genus Emergomyces would be listed for the cross-reaction against *P. ceti* [6,8]; however, the present study completely disproves the serological cross-reaction between coccidioidomycosis and PCM-C. Thus, we hypothesize that the cross-reactions caused by the related fungal species to *Coccidioides* spp., such as *Uncinocarpus reesii*, *Chrysosporium queenslandicum*, and *Chrysosporium* spp., recorded in the Far East are ignored. The antibodies against *H. capsutlatum* and *A. kalrae* reacted positively to all the antigens used in the present study, but their influences on the sero-positiveness against *P. ceti* fungal cells were already denied because of the limited genotype on *H. capsulatum* and no record of isolation from marine environments on *A. kalrae* [5,6].

In conclusion, the true serological cross-reaction of the antibody against *P. ceti* is limited between paracoccidioidomycosis and PCM-C; the method for evaluation of the antibody against PCM-C is reliable, and there is potential for epidemiological study.

## Figures and Tables

**Figure 1 microorganisms-11-02428-f001:**
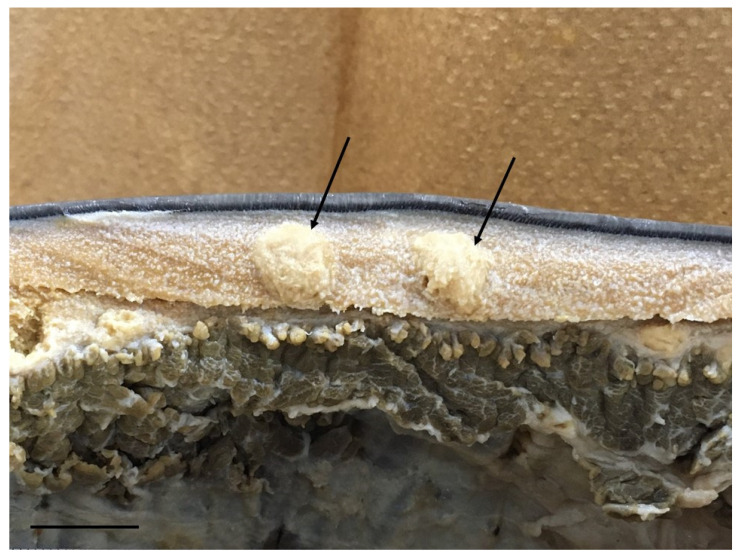
The foci in the subcutaneous connective tissue of the first Japanese case of bottlenose dolphin with PCM-C fixed with 10% formalin (arrows). The bar indicates 20 mm.

**Figure 2 microorganisms-11-02428-f002:**
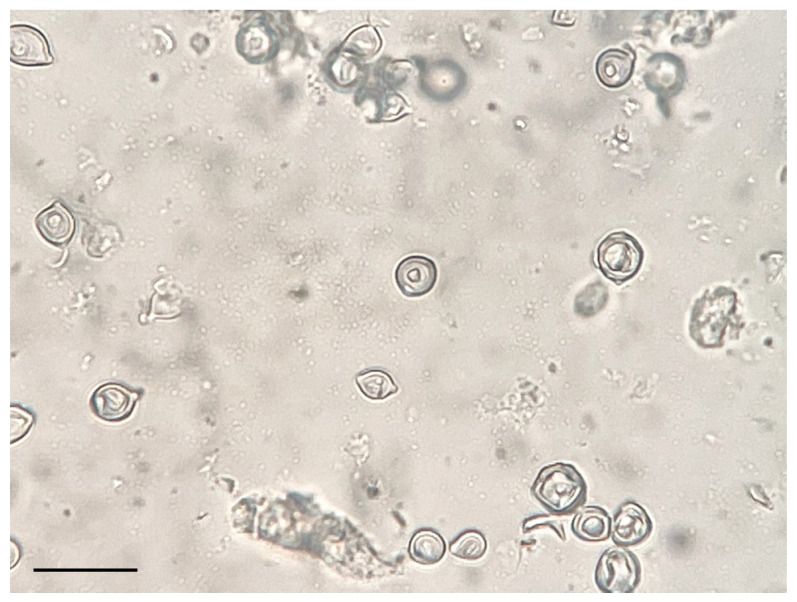
The fungal cell solution consisted of round yeast cells with a small debris of cetacean tissue. The bar indicates 20 μm.

**Figure 3 microorganisms-11-02428-f003:**
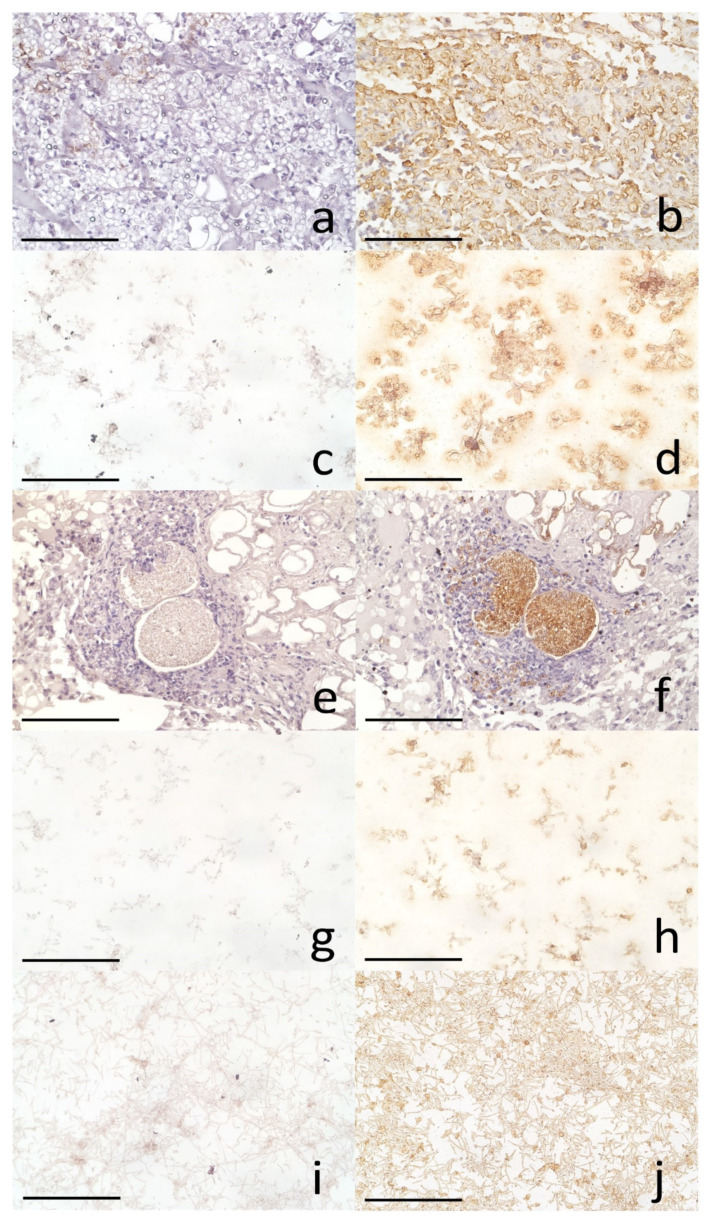
Immunohistochemical reactions of *Paracoccidioides ceti* yeast cells ((**a**) negative, (**b**) positive example using a 1000-fold diluted rabbit serum derived from the rabbit experimentally inoculated with formalin-fixed *P. ceti* cells in the first case of paracoccidioidomycosis ceti in a bottle nose dolphin, 400× magnification), *P. brasiliensis* yeast cells ((**c**) negative, (**d**) positive example using a 1000-fold diluted rabbit serum derived from the rabbit experimentally inoculated with yeast cells of *P. brasiliensis*, 400× magnification), *C. posadasii* ((**e**) negative, (**f**) positive example using a 1000-fold diluted cetacean serum from a Dall’s porpoise which reacted positively to *C. posadasii* in our previous report [8], 200× magnification), *Histoplasma cupslatum* ((**g**) negative, (**h**) positive example using 1000-fold diluted pooled sera from five mice experimentally infected with IFM41329 [5], 400× magnification), and *Arthrographis karlae* ((**i**) negative, (**j**) positive example using 1000-fold diluted pooled sera from four mice experimentally infected with IFM55165 [6], 400× magnification). The bars indicate 20 μm, except for (**e**,**f**), which measures 50 µm.

**Table 1 microorganisms-11-02428-t001:** List of antibodies for positive controls.

Antibody	Origin of the Serum	Remarks
1. *Paracoccidioides cetii*	Rabbit	Experimentally produced by Eurofingenomics, Co., Ltd.

2. *Paracoccidioides brasiliensis*	Rabbit	Experimentally infected rabbit
3. *Coccidioides* spp.	Dolphin	Dall’s porpoise [8]
4. *Histoplasma capsulatum*	Mice	Pooled sera from five mice experimentally infected with IFM 41,329 [5]

5. *Arthrographis karlae*	Mice	Pooled sera from four mice experimentally infected with IFM 55,165 [6]


**Table 2 microorganisms-11-02428-t002:** List of antigens used in this study.

Antigen	Remarks
1. *Paracoccidioides cetii*	SUM [10]
2. *Paracoccidioides brasiliensis*	Pb-18 [5]
3. *Coccidioides posadasii*	IFM 4935 [5]
4. *Histoplasma capsulatum*	IMT/HC128 [5]
5. *Arthrographis karlae*	IFM55165 [5]
	(Cutaneous lesion of a cat)

**Table 3 microorganisms-11-02428-t003:** Immune reactions of the fungal cells versus antibodies.

Antibodies	Source of the Serum	PCT	CP	PB	HP	AK
*P. cetii*	Rabbit	+	−	+	−	−
*P. brasiliensis*	Rabbit	+	−	+	−	−
*C. posadasii*	Dolphin *	−	+	−	−	−
*H. capsulatum*	Mice	+	+	+	+	+
*A. karlae*	Mice	+	+	+	+	+

* Sera from a dolphin (Dall’s porpoise) in our previous study. Abbreviations: PCT (*P. ceti*); CP (*C. posadasii*); PB (*P. brasiliensis*); HC (*H. capsulatum*); AK (*A. kalrae*).

## Data Availability

Not applicable.

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
