# Peer review of "Re-Evaluation of the Cross-Reactions of the Antibody against the Causative Agent for Paracoccidioidomycosis Ceti; *Paracoccidioides ceti* and the Related Fungal Species"

_microorganisms, 2023, doi:10.3390/microorganisms11102428_

Round 1

Reviewer 1 Report

microorganisms-2558023-peer-review-v1

The present paper is interesting and has clear practical application in addition to the scientific value. In my opinion this work can be suggested for publication, however, appropriate corrections and upgrades need to be taken into consideration by the authors. Maybe help from more experienced colleagues will be a good option in critical reading and correction of the manuscript.

Abstract needs to be presented better, with some specific results recorded in current study.

Please, be sure that you have intervals between words and provided references.

Centrifugations need to be as "xg" and not as "rpm".

In Material and methods, under 2.1.: by or ly?

For all suppliers of material and equipment, please, provide the name of the company, city, state (in case of federal country) in abbreviated way and name of the country. In following occasions, only name of the company will be sufficient.

Table 3 needs to be formatted.

Discussion is very preliminary. maybe authors will consider extending discussion section.

References are not into the recommended by the journal format. Please, format them.

Reviewer 2 Report

The authors attempted to re-evaluate cross-reactions of anti-Paracoccidioidomycosis ceti, Paracoccidioides ceti and related species of fungi, mostly of marine origin in rabbits. The methodology is fine (although crude methods have been used), however the discussion part has been neglected to convey a scientific meaning of the work to the audience. I would recommend merging results and discussion and also shortening the methodology as much has already been described before. The authors should discuss the scientific significance of this work in the discussion section.

English is fine, only minor corrections...
